# High-Accuracy Guide Star Catalogue Generation with a Machine Learning Classification Algorithm

**DOI:** 10.3390/s21082647

**Published:** 2021-04-09

**Authors:** Jianming Zhang, Junxiang Lian, Zhaoxiang Yi, Shuwang Yang, Ying Shan

**Affiliations:** MOE Key Laboratory of TianQin Mission, TianQin Research Center for Gravitational Physics & School of Physics and Astronomy, Frontiers Science Center for TianQin, CNSA Research Center for Gravitational Waves, Zhuhai Campus, Sun Yat-Sen University, Zhuhai 519082, China; zhangjm66@mail2.sysu.edu.cn (J.Z.); yizhx6@mail.sysu.edu.cn (Z.Y.); yangshw8@mail2.sysu.edu.cn (S.Y.); shany5@mail2.sysu.edu.cn (Y.S.)

**Keywords:** star sensor, guide star catalogue, machine learning

## Abstract

In order to detect gravitational waves and characterise their sources, three laser links were constructed with three identical satellites, such that interferometric measurements for scientific experiments can be carried out. The attitude of the spacecraft in the initial phase of laser link docking is provided by a star sensor (SSR) onboard the satellite. If the attitude measurement capacity of the SSR is improved, the efficiency of establishing laser linking can be elevated. An important technology for satellite attitude determination using SSRs is star identification. At present, a guide star catalogue (GSC) is the only basis for realising this. Hence, a method for improving the GSC, in terms of storage, completeness, and uniformity, is studied in this paper. First, the relationship between star numbers in the field of view (FOV) of a staring SSR, together with the noise equivalent angle (NEA) of the SSR—which determines the accuracy of the SSR—is discussed. Then, according to the relationship between the number of stars (NOS) in the FOV, the brightness of the stars, and the size of the FOV, two constraints are used to select stars in the SAO GSC. Finally, the performance of the GSCs generated by Decision Trees (DC), K-Nearest Neighbours (KNN), Support Vector Machine (SVM), the Magnitude Filter Method (MFM), Gradient Boosting (GB), a Neural Network (NN), Random Forest (RF), and Stochastic Gradient Descent (SGD) is assessed. The results show that the GSC generated by the KNN method is better than those of other methods, in terms of storage, uniformity, and completeness. The KNN-generated GSC is suitable for high-accuracy spacecraft applications, such as gravitational detection satellites.

## 1. Introduction

A gravitational wave (GW) signal was first observed by LIGO, which successfully confirmed the prediction of Einstein’s general relativity (GR) [1]. The signal sweeps upwards in frequency, from 35 to 250 Hz, with a peak gravitational wave strain of $1.0\times 10^{-21}$ [2]. However, gravitational waves have important astronomical sources in the millihertz (mHz) range (i.e., 0.1–100 mHz) [3]. In order to detect the important gravitational waves at low frequencies, it is necessary to go into space. The proposed space-borne detection plans, LISA [4], TianQin [3], and Taiji [5], use laser interferometric systems (see Figure 1) constructed using three identical satellites, each carrying two telescopes for laser pointing. Nevertheless, the interferometric measurements for the experiment are only possible once the three laser links between the three identical spacecrafts are acquired. The attitude of the spacecraft is measured by the star sensor (SSR) and provides an initial rough range for the laser link scanning; in other words, the attitude measurement capacity of the SSR can influence the efficiency of the link docking. Hence, the method used to elevate the measurement capacity of the SSR is important for laser acquisition. This paper aims to improve the generation method of the star catalogue, in order to improve the measurement capacity of the SSR.

In astronomy, star data in the celestial sphere are compiled into a catalogue, according to different requirements. The catalogue is called a star catalogue (or star catalog). In celestial navigation, a GSC (guide star catalogue) is the unique basis for a star sensor (SSR), which facilitates star identification [6]. An SSR is key in celestial navigation. Compared with sun sensors, earth sensors, magnetometers, horizon sensors, and other common attitude measurement sensors, SSRs have a higher attitude measurement accuracy, can realise navigation independently, and have a strong anti-interference ability. At present, they are the most important attitude sensors used in satellites and other spacecraft. The ability of an SSR to determine the attitude depends on the existence of an excellent navigation star catalogue. Generally, a preferable star catalogue for navigation is characterised by a lower number of guide stars, a higher completeness, and a much more uniform distribution of the celestial sphere [7].

The most frequently used strategy for selecting guide stars for an SSR is the magnitude filtering method (MFM). Considering the starlight sensitivity of the SSR, a fixed minimum visual magnitude threshold (VMT) is set, and a maximum, “VMT_1”. Stars that are brighter than or equal to the VMT and dimmer than or equal to VMT_1 in the catalogue are chosen as guide stars [8]. The correct identification of a star atlas in a local celestial area (<100 square degrees) requires at least three guide stars among the observation stars, and at least five guide stars for the entire celestial star identification process [9]. After processing by MFM, the stars are still unevenly distributed in the celestial sphere, and according to statistics, the maximum is about 10 times larger than the minimum. If we set the value of VMT_1 too small (in other words, the maximum value of the visual magnitude of the star is too small), the GSC will be incomplete, leading to a lack of stars in some directions (called holes). This will lead to a reduction in the success rate of star identification and measurement misalignment [10,11]. However, if the value of VMT_1 is too large, the star number also increases, leading to redundancy in the GSC. As a result, the efficiency of star identification may be reduced. Completeness, uniformity, and redundancy are the three most important characteristics of a GSC. It is necessary to probe the relationships among them when selecting a GSC.

There are many studies that have considered the uniformity, completeness, and redundancy of star catalogues. Farshad et al. [12] aimed to ensure the uniformity of the database for an SSR, and a weighted K-means clustering method (GWKM) was proposed, on the basis of SKM [13] and GKM [14]. The results of this research showed that the number of stars was reduced while promoting uniformity, improving the refresh rate of star identification and the efficiency of SSRs.

Ivan et al. [15] pointed out that the efficiency of star identification depends on the quality of the GSC. They decreased the number of stars (NOS) in dense areas, while the sparse areas had no change. The average NOS in the FOV was, thus, decreased. Finally, the distribution of the stars became more symmetrical and bell-shaped. Additionally, instrumental stellar magnitude estimation and a lower bound evaluation method were also discussed. The onboard star catalogue was sufficient for middle-precision SSRs; however, it was not satisfactory for high-accuracy SSRs.

When the celestial sphere is divided geometrically, the most representative method is the inscribed cube algorithm [16]. An inscribed cube is used to evenly divide the celestial sphere into six parts, where each of the six parts is then divided into N×N smaller parts. Finally, an evenly and non-overlapping partition of the guide star selection is realised. However, Li et al. [7] pointed out that the solid angles are different in size for each sub-block, especially when the optical axis of the SSR is around the celestial pole. They proposed a novel method to generate a “quasi-uniform” star catalogue, in order to solve the differences in solid angles. There were 2664 sub-blocks that made up the whole celestial sphere in total, where the reference was the solid angle corresponding to $4^{\circ }\times 4^{\circ }$. According to the statistical results, the probability of at least three guide stars being in the FOV was more than 99.9%.

According to Zhang et al. [17], except for neural networks, most methods for recognising a star atlas need a star catalogue. At present, MFM is the most popular method used to generate star catalogues. However, there are holes and redundancy problems when using the MFM. Zhang proposed a Support Vector Machine (SVM) to generate a star catalogue. Their results showed that the SVM has high flexibility. This is a successful case of applying a classification algorithm for the generation of a GSC. Sun et al. [18] combined NOS with Boltzmann entropy [19] within a circular region as a feature vector and then used an SVM to select stars to generate an even GSC. Through an experimental comparison, the SVM algorithm was shown to be much better than the MFM, in terms of the distribution uniformity. On the basis of Sun’s method, Liu et al. [20] used a sphere spiral algorithm to create uniform sampling data and then used the SVM to select the guide star. Finally, defined global and local criteria were used to assess the performance of the created GSC. By comparing a self-organising algorithm (S-OA), the MFM, and a magnitude weighted method (MWM), they found that the proposed method was optimal and had strong adaptability while preserving uniformity.

In summary, the MFM can be used to obtain a streamlined GSC. However, there are still holes and redundancy problems. To solve this problem, many methods have been devised. The SVM and other algorithms, as detailed above, have been introduced to address the uniformity and redundancy of the GSC. However, the redundancy and completeness are still difficult to make compatible. Depending on the nature of the mission, one of them generally must be selectively sacrificed. At present, detection missions are more complex. SSRs have also made comprehensive progress. More attention has been paid to the GSCs for multi-field SSRs and high dynamic SSRs, and the generation of an appropriate GSC is still very important. Most of the work has focused on the completeness, uniformity, and redundancy of the GSC. However, the attitude determination accuracy of SSRs has become increasingly important in space missions, while accuracy criteria have not been considered in most GSC generation methods. The SVM classification algorithm has a good effect on the generation of GSCs. In the machine learning field, there exist many classification algorithms that can achieve the same or an even better classification effects than the SVM. However, few works have discussed the advantages, disadvantages, and adaptability of these algorithms for the generation of GSCs.

This paper is devoted to creating a uniform and complete star catalogue by taking advantage of the K-Nearest Neighbours classification algorithm. The NOS in the FOV of an SSR is considered, in order to ensure sufficient accuracy for an SSR. It is chosen as an important star selection criterion. Firstly, we used datasets, after the MFM and double star processing, as the celestial star data. Subsequently, a sequential scanning of the entire celestial sphere was carried out, and training and validation datasets were obtained. Then, we use thed samples to identify the performances of the classification algorithms. We found that the K-Nearest Neighbours (KNN—82.0%) and Decision Tree (DC—82.0%) performed the best. Finally, we used SGD (Stochastic Gradient Descent), KNN, DC, NN (Neural Network), RF (Random Forest), SVM, and GB (Gradient Boosting) to generate star catalogues, and a test considering 10,000 random boresight directions of the entire celestial sphere was conducted. On the basis of statistical analysis, the KNN classification algorithm was found to be far superior to the MFM and SVM—which are normally used to generate GSCs—in terms of completeness, uniformity, and redundancy. Furthermore, the star catalogue can guarantee an SSR accuracy within 1 arc sec.

## 2. Star Numbers and SSR Accuracy

The method of establishing a correspondence between measured stars in the FOV and an onboard GSC is called star pattern recognition. According to Liebe [10,11,21], the NOS in the FOV of an SSR is connected with the accuracy of star identification. For an SSR, in order to explore the relationship between the NOS in FOV and accuracy, the first thing to consider is the average NOS in the FOV. It is considered that the GSC is sufficiently evenly spread over the entire celestial sphere. The relationship between them is as follow:(1)$$\begin{array}{c} \begin{array}{c} N_{FOV}=6.57\times e^{\left(1.08\cdot M\right)}\times \frac{1-\cos(\frac{A}{2})}{2}. \end{array} \end{array}$$
where *A* represents the circular FOV which is *A* deg wide. The corresponding part of the sky covered by the FOV is (1−cos(A/2))/2, which is used as one part of Equation (Equation 1). *M* is the magnitude sensitivity limit; this is the NOS that are brighter than a set magnitude. The value of *M* is evaluated by sketching the correlation between the magnitude and the NOS which are brighter than the set magnitude. According to [11], $N_{FOV}=6.57\cdot e^{\left(1.08\cdot M\right)}$, which is another part of Equation (Equation 1). Finally, the average NOS in the FOV is given by Equation (Equation 1).

The relationships between the average NOS in FOV, the FOV size, and the visual magnitude are shown in Figure 2. As we can see, the average NOS in the FOV increases with the increase of the size of FOV and the value of *VMT,* within a certain range. The lower figure in Figure 2 shows that, when *VMT* = 5 and FOV=10 or VMT=6 and FOV=10, about three stars and eight stars, respectively, appear on average in the FOV. The attitude accuracy is far less than 1 arc sec. To satisfy the accuracy requirements, and referring to Figure 2, there must be at least 18 stars in the FOV. Thus, an FOV of $12^{\circ }$ and a VMT of 8 were chosen as the parameters of the SSR.

The high frequency error NEA (noise equivalent angle) is a random measurement error. It can be regarded as the capacity of an SSR to reappear at the identical attitude [11]. NEA errors are caused by device noise, stray light interference (e.g., from the sun, earth, and bright stars), timing jitter, model error (stellar spectral distribution), non-uniformity of tracking magnitude, and so on. In this paper, we guarantee the NEA to meet the theoretical attitude accuracy of the SSR. The NEA can represent the accuracy of the SSR, and a few parameters are used to estimate the NEA. The cross-boresight (not the optical axis) NEA [11] is as follow:(2)$$\begin{array}{c} \begin{array}{c} \sigma_{yaw,pitch}=\frac{A_{FOV}\times \sigma_{centroid}}{N_{pixel}\times \sqrt{N_{star}}}. \end{array} \end{array}$$
where $A_{FOV}$ is the FOV of this SSR (generally $1^{\circ }$ to $24^{\circ }$; $12^{\circ }$ is considered in this paper) and $\sigma_{centroid}$ is the average centroiding accuracy. At present, the star image points on the sensitive surface of the image sensor are defocused, in order to spread to more pixels to obtain higher star centroid localisation accuracy. The diameter of the diffusion circle is generally 3 to 5 pixels, typically ranging from 0.01 to 0.5. We assume the $\sigma_{centroid}$ is 0.1 pixel. $N_{pixel}$ represents the number of pixels across the focal plane, typically ranging from 256 to 1024 (here, it is taken as 1024). $N_{star}$ is the NOS in this FOV, as Figure 2 shows. In this paper, $N_{star}$ is a vital parameter to create a star catalogue, as we are committed to creating a star catalogue that can provide high-accuracy attitude information.

The boresight (optical axis) NEA of an SSR can be estimated as follows: First, suppose that the quadratic focal plane of this SSR is N×N pixels. The average distance from the star mapped to the focal plane to the center of it is calculated as:(3)$$\begin{array}{c} \begin{array}{c} \int_{-N/2}^{N/2}\int_{-N/2}^{N/2}\sqrt{x^{2}+y^{2}}dxdy=0.3825N. \end{array} \end{array}$$

As the optical axis is the roll axis, there is an uncertain spatial range on the roll axis. We set it as $\sigma_{centroid}$. The accuracy of a single star is derived from its geometric relationship with $\sigma_{centroid}$:(4)$$\begin{array}{c} \begin{array}{c} E_{roll\;single\;star}=atan\frac{\sigma_{centroid}}{0.3825\times N_{pixel}}. \end{array} \end{array}$$

Statistically, the measurement of the centroid positions of multiple stars in a frame is independent and uncorrelated, and thus, this measurement is used to improve the entire attitude accuracy. Hence, the parameter $\sqrt{N_{star}}$ is introduced. Finally, we obtain the roll (optical axis) accuracy for the entire attitude estimate as follows:(5)$$\begin{array}{c} \begin{array}{c} E_{roll}=atan\left(\frac{\sigma_{centroid}}{0.3825\times N_{pixel}}\right)\times \frac{1}{\sqrt{N_{star}}}\times \frac{180}{\pi }. \end{array} \end{array}$$

Figure 3 shows the relationship between the accuracy of three axes and the NOS in the FOV. As the focal length of the SSR is much larger than that of the image sensor in the optical system, the attitude accuracy error of the SSR in the direction of the optical axis (usually the “roll” axis) is about 6–16 times larger than that in the two directions on the focal plane (“yaw” and “pitch”), as shown in Figure 3.

## 3. Data and Pre-Processing

Full-sky star catalogues commonly used for satellite attitude determination are listed in Table 1. There are many stars in the catalogue, but not all of them will be used in celestial navigation. Usually, it is necessary to refer to the nature of the specific tasks, in order to obtain the stars satisfying the mission from the basic catalogue, then to construct the on-board navigation catalogue.

The Smithsonian Astrophysical Observatory Star Catalogue was compiled, in 1966, from the Smithsonian Observatory and other catalogues. In 1979, the radian information of equatorial co-ordinates and the cross-validation information of SAO/HD/DM/GC were added. These data have been updated, according to the latest observation information in 1984, based on J1950.0. The recent catalogue was released in 1990, and was based on J2000.0. The star table contains 258,997 celestial bodies with very accurate positions and motions [22]. For satellite attitude calculation by an SSR, the position accuracy factor is far greater than the magnitude accuracy [21]. The position accuracy of the SAO Catalogue can reach $10^{-8}$, and the magnitude accuracy is $10^{-1}$, so this meets the requirements of attitude calculation well. The catalogue also covers the whole sky, which means that it has a high completeness. For these reasons, we chose it as our basic star catalogue.

The data of the SAO includes 258,997 rows of star information, where each row includes 204 bytes, representing 57 types of information about the stars. What we are most concerned with is the visual magnitude, the celestial right ascension of J2000.0, and the celestial declination of J2000.0. Due to the fact that stars are far away from us, the field angles of stars are much less than 1 arc second, when observed from earth. Stars with a very close sight (called double stars, not binary stars, in astronomy) cannot be distinguished from each other on the imaging plane, and will interfere with the star identification process. Therefore, the general star identification algorithm is not appropriate for double stars [23]. Double stars are deleted directly, in general. However, when the NOS in the FOV is very small, each one is vital for the identification. In view of this, we combined such stars into one, in order to ensure the completeness of the GSC.

Suppose $m_{1}$ and $m_{2}$ are the double star’s magnitudes, and the right ascension and declination are $\alpha_{1}$, $\alpha_{2}$, and $\beta_{1}$, $\beta_{2}$, respectively. The direction vectors $r_{1}$ and $r_{2}$ of the two stars are calculated from their right ascension and declination, and the visual magnitude of the combined star is $m_{0}$. The combined brightness is *F* and the direction vector is $r_{0}$. The optical flux density can represent the brightness of stars, and the luminance ratio of the double star is expressed by Equation (Equation 6). Then, we combine Equations (Equation 6) and (Equation 7) and obtain Equation (Equation 8). Finally, the brightness of composed star is expressed by Equation (Equation 9).
(6)$$\begin{array}{c} \begin{array}{c} \frac{F_{1}}{F_{2}}=e^{(m_{2}-m_{1})/2.5}, \end{array} \end{array}$$
(7)$$\begin{array}{c} \begin{array}{c} F=F_{1}+F_{2}, \end{array} \end{array}$$
(8)$$\begin{array}{c} \begin{array}{c} \frac{F}{F_{2}}=\frac{F_{1}+F_{2}}{F_{2}}e^{(m_{2}-m_{0})/2.5}, \end{array} \end{array}$$
(9)$$\begin{array}{c} \begin{array}{c} m=m_{2}-2.5\times \ln\left(1+\frac{F_{1}}{F_{2}}\right)=m_{2}-2.5\times \ln\left(1+e^{(m_{2}-m_{1})/2.5}\right). \end{array} \end{array}$$

Assume that the angular distances among this combined star with the double star are $\phi_{1}$ and $\phi_{2}$. The angular distance between these two double stars is $\phi_{0}$. Then, we obtain Equation (Equation 10). As $\phi_{0}$ is determined through the calculation of their right ascension and declination, we can obtain the values of $\phi_{1}$ and $\phi_{2}$. The direction vector, $r_{0}$, of the composed star is obtained by Equation (Equation 11).
(10)$$\begin{array}{c} \begin{array}{c} F_{1}\phi_{1}=F_{2}\phi_{2},\\ \phi_{0}=\phi_{1}+\phi_{2}=\phi_{1}\left(1+\frac{F_{1}}{F_{2}}\right)=\phi_{1}\left(1+e^{(m_{2}-m_{1})/2.5}\right), \end{array} \end{array}$$
(11)$$\begin{array}{c} \begin{array}{c} r_{0}sin\phi_{0}=r_{1}sin\phi_{1}+r_{2}\phi_{2}\approx r_{1}\phi_{1}+r_{2}\phi_{2},\\ r_{0}=\left(r_{1}\phi_{1}+r_{2}\phi_{2}\right)/sin\phi_{0}. \end{array} \end{array}$$

According to Zhang [23], in the process of star centroid extraction, the threshold of binarisation is *T* and the double star must satisfy the minimal distance *d*:(12)$$\begin{array}{c} \begin{array}{c} T=2B\times e^{\frac{-{\left(d/2\right)}^{2}}{2\sigma^{2}}}, \end{array} \end{array}$$
where σ=1, B=255, and *B* represents the brightness of a star. For an SSR with a $12^{\circ }$ FOV and 1024×1024 pixels, *d* is determined as 4 pixels, which is about $0.047^{\circ }$. The basic star catalogue, SAO, was reduced to a new star catalogue called “MFM” by setting VMT=8 and processing the double stars. Figure 4 and Figure 5 show the distribution and density of the SAO and MFM star catalogues, respectively.

Figure 4a and Figure 5a exhibit the distribution density of the GSC in the celestial sphere, in which the colour in the colour bar represents the magnitude of the density, and the colour above the colour bar represents the NOS in this region. It is shown that the SAO catalogue’s distribution is more dense than that of the MFM catalogue, and the colour in Figure 4a is darker (greener). Based on Figure 4a and Figure 5a, we can make an intuitive conclusion that the MFM catalogue has advantages, in terms of data capacity; namely, redundancy. Both SAO and MFM have high completeness.

Figure 4b,c and Figure 5b,c show the statistics of the NOS distributed over the entire celestial sphere. In the direction of right ascension and declination, they are equally divided into 50 parts, and the corresponding NOS in each range is counted. In these figures, on the right ascension, stars are most concentrated at about $100^{\circ }$ and $300^{\circ }$; on the declination, stars are most concentrated at about $\pm 40^{\circ }$. The distribution of stars in the GSC obtained by the MFM is similar to that of the GSC SAO, but the NOS is greatly reduced.

Figure 4d and Figure 5d show that the nuclear density line of the GSC stars in the celestial sphere. The nuclear density curve more clearly shows the distribution of stars in the sky. The right colour scale shows the size of the nuclear density. The larger the scale is, the higher the nuclear density is. That is to say, stars are densely distributed in this part of the celestial sphere. Comparing the numerical values and the nuclear density curves, the distribution of SAO is denser and the MFM is relatively more uniform.

## 4. Generation of the KNN GSC

In the above two sections, we obtained the FOV size and threshold sensitivity of the SSR and processed the double stars in the SAO. In this section, we mainly describe using the KNN to generate the KNN GSC. First, we introduce machine learning classification algorithms. Then, we introduce the principle of the KNN algorithm. Finally, we introduce the specific implementation steps of using the KNN generating the GSC.

### 4.1. Machine Learning and Classification Algorithms

Machine learning (ML) is a form of artificial intelligence (AI) that is mainly used to study how the performance of specific algorithms can be improved in empirical learning [24]. The purpose of ML is to realise AI. Classification in ML is a supervised learning method (SLM), with statistical learning theory serving as the theoretical basis. An SLM builds a mathematical model from a set of data (known as training data) that contains both the input and the desired outputs [25]. The main difference between SLMs and unsupervised learning is whether the dataset is labelled. Classification algorithms in ML can be regarded as a form of “pattern recognition”. The classification algorithms use input training data to predict the likelihood that subsequent data will fall into one of the pre-determined categories; that is to say that the data follow the same pattern.

### 4.2. KNN Classification Algorithm

In this study, we used classification algorithms to generate GSCs. The used classification algorithms were from the scikit-learn machine learning tool [26], which contains KNN, RF, NN, DC, SVM, SGD, and GB, among others. In machine learning classification, the classifier learns from labelled data (usually named training samples). After the classifier understands the data, the algorithm determines which label to assign to the new data (usually named test samples) by associating the schema with the (unlabelled) new data.

Classification determines the category of objects, based on one or more independent variables. The KNN was determined as the optimal classifier for our star catalog selection. For classification, when the dataset has little or no prior knowledge of the distribution, the KNN algorithm has the most advantages. Furthermore, KNN classification is one of the most fundamental and simple methods. KNN is a non-parametric algorithm [27] and is a famous representative of “lazy learning”. It classifies new data according to similarity (e.g., a distance function with weight).

For classification, unlike other classification algorithms, KNN does not build a general internal model through the training dataset; rather, it stores the data for the nearest voting step, such that the algorithm is relatively simple. In scikit-learn, there are two different nearest neighbour classifiers: “KNeighboursClassifier” (KNC) and “RadiusNeighboursClassifier” (RNC). We adopted the most common technique, KNC, where *k* represents the number of nearest neighbours that are distributed around the samples to be classified. The other classifier, RNC, is given a fixed radius and counts the number of nearest neighbours in its region. The KNN algorithm is highly dependent on the data. A larger *k* can suppress noise in the data, but will increase the complexity of the algorithm. If *k* is too small, the noise will be amplified and overfitting may occur. The optimal *k* value is usually selected by cross-validation (see Section 4.3).

We used a variety of machine learning classification algorithms to generate GSCs, where the steps of generating the GSCs were almost the same. We briefly introduce the best (KNN) in the following. The KNN classification algorithm can be summarised in the following steps:1.Assume that there are *n* samples in the training set, $\left\{x_{1},x_{2},...,x_{n}\right\}$, which belong to *c* categories $\left\{y_{1},...,y_{c}\right\}$.2.Calculate the distance (usually the Euclidean distance) between the test data $x_{k}$ and each training datum. The distance in N-dimensional space is defined as $D\left(x,x_{k}\right):=\sqrt{\sum_{i=1}^{n}{\left(x_{i}-x_{k}\right)}^{2}}$, where *x* is the sample in the training dataset.3.Sort the data, according to the increasing relation of distance.4.Select the *k* stars with the smallest distance.5.Determine the frequency of occurrence of the category of the first *k* stars.6.The most frequent category in the first *k* stars is returned as the prediction classification result. *k* is the number of nearest stars.

In Figure 6, when k=3, we can see that there is one guide star (the blue star) and a non-guide star (the red star) in the circle; thus, we cannot classify the stars. When k=5, there are three guide stars and two non-guide stars, so we decide to classify the star (the green star) as a guide star.

### 4.3. Generating GSC by KNN

Star catalogue generation is regarded as a binary classification problem. There are three features: right ascension, declination, and star magnitude. The features of the stars to be classified are compared with the features of the stars in the training dataset. If the first *k* data samples in the training dataset are found to be the most similar, the category corresponding to the test dataset is the one with the most frequent occurrence. The classification principle is simply expressed in Figure 6, and we assume that the stars are distributed in Figure 6, according to the features.

In this paper, *k* is an important parameter of the classifier. Through cross-validation, we start by selecting a smaller value of *k*. Then, we increase the value, calculate the variance of the validation set, and finally find a more appropriate value of *k*. Here, we determined that the optimal value of *k* was 15. Another important parameter of the KNN classifier in scikit-learn is the weight. When it is selected as “uniform”, each neighbour is assigned a uniform weight; if the value of the weight is selected as “distance”, it assigns a weight proportional to the inverse of the distance from the query point. We chose “distance”, in order to give closer neighbours more weight.

The specific steps of using the KNN classification algorithm to generate a GSC are shown in Figure 7. The SAO catalogue processing, feature extraction, threshold filtering, and accuracy analysis were introduced in the above sections. We chose “n”, the brightest stars in each simulated FOV, as guide stars and labelled them as “1”, meaning that we first traversed the entire sky, according to the right ascension and declination. We took half of the FOV to simulate SSR imaging. Later, we chose “m”, the brighter stars, as non-guide stars and labelled them as “−1”. In other words, according to the brightness of the stars, take “m” stars as non-guide stars, after “n” brightest stars. The stars “n” and “m”, taken together, formed a sample set with labels. This ensured that stars could be selected as guide stars in sparse celestial regions.

We obtained the training and test samples using the MFM star catalogue. Training samples were used to train the corresponding classifier. The classifier was then used to classify samples. The training samples and test samples were divided according to a proportion of 3:1. After label comparison, the accuracy of the classification algorithm was determined. The accuracy, $E_{c}$, is the ratio of the correct classification in the star sample C to the total number of samples S, $E_{c}=\left(C/S\right)\times 100\%$, as shown in Figure 8. KNN (82.0%) was found to have the best performance.

In the final steps, shown in Figure 7, the average NOS in the FOV was determined, and the corresponding accuracy of the SSR was calculated to determine whether the selection of “m” and “n” was appropriate. First, smaller values of “m” and “n” were determined. The values of “m” and “n” were then increased, until the accuracy was at least 1 arc sec. This is a time-consuming step. Finally, through adjustment, the appropriate values of “m = 5” and “n = 12” were obtained.

## 5. Evaluation of the KNN GSC

In order to further evaluate the uniformity, completeness, and redundancy of the KNN GSC and the GSCs generated by other machine learning classification algorithms, we created 10,000 random directions, with a $12^{\circ }$ FOV, $N_{pixel}=1024$, and VMT=8. The statistics are shown in Table 2. As can be seen in Figure 9 and Table 2, the maximum and mean of the stars in the FOV of KNN (14,344.73) were better than those of the DC (16,353.18) and other GSCs, and the Std (standard deviation) of KNN (15.62) was also the best among those classification algorithms. The uniformity of the star catalogue shown in Figure 9 is much better than in Figure 5. Hence, the proposed KNN machine learning classification algorithm can generate a uniform, complete, and highly accurate GSC.

For an SSR’s star catalogue construction, the commonly used method is the SVM classification algorithm. In our experiment, we found that the SVM was not the best for the process of star catalogue generation when using machine learning classification algorithms. The storage of the catalogue, the average NOS, the Std of stars, and the accuracy of an SSR with the NOS of the SVM and KNN are shown in Table 2. These represent the star catalogue’s quality. The catalogue generated by the KNN was better than the catalogue generated by the SVM, in all of the above aspects. In other words, we introduced a new classification algorithm for selecting a GSC, which is better than the most commonly used classification method (i.e., SVM).

## 6. Discussion

In this paper, we considered the generation of a GSC as a classification problem and used a variety of machine learning classification algorithms to generate GSCs. Overall, the KNN was found to be the best classifier. All of the methods generated a GSC using the same steps, the only difference being the classifier. The KNN algorithm is simple and easy to implement; there is no need to estimate parameters, and it is suitable for rare event classification; however, it displays poor interpretability (a common problem of machine learning algorithms) and is highly dependent on data—wrong data may directly lead to inaccurate predictions.

The reasons why the KNN was found to be the best method to generate a GSC herein are as follows: (1) The sample set was classified with more crossed or overlapping class domains, as the KNN method mainly depends on a limited number of neighbouring samples, rather than distinguishing class domains. Most of the star feature samples are overlapped or crossed, which is suitable for a KNN. (2) Compared with big data, the samples used to generate a GSC are relatively few, and the KNN algorithm is suitable for small samples.

For different classification problems, machine learning classification algorithms have different performance; further, KNN algorithms have the disadvantage of unexplainability. These two points are recognised in the field of machine learning and need further research. However, the KNN algorithm is exactly suitable for the guide star generation problem studied in this paper.

The uniformity of the KNN GSC needs to be further improved. Somayehee et al. [12] created a uniform GSC using a novel method of weighted k-means clustering (a machine learning algorithm) with geodesic criteria. However, the main work of this paper was obtaining a machine learning classification that can satisfy high-precision attitude measurement accuracy, for which we identified the KNN algorithm, based on uniformity, completeness, and redundancy standards, which were based on a more comprehensive analysis. Further improvement in GSC uniformity will be pursued in future research.

## 7. Conclusions

Differing from most other works, we focused on generating a uniform and complete GSC, while ensuring that it can provide an accuracy of at least 1 arc second for an SSR. First, we used the MFM method to create a basic GSC. Then, we constructed the training and validation datasets (with a ratio of 3:1) and compared the accuracy of 7 different machine learning classification algorithms, in order to determine the best method. KNN and DC were the two best, both of which had an accuracy of 82.0%. After comparison, KNN was determined as the best algorithm for generating a GSC. Three criteria—accuracy, uniformity, and completeness—were considered. Accuracy was the basic criterion. After 10,000 Monte Carlo random experiments, a smaller capacity was obtained and a more uniform GSC was generated by the KNN algorithm. The redundancy of the SSR was also reduced. This method was shown to be much better than the commonly used SVM classification methods. KNN algorithm is introduced for the first time in the construction of a navigation star catalogue. Moreover, the mean NOS in the FOV was 44.73 and the minimum NOS was 15. Both achieved an attitude accuracy of 1 arc second. There were 16,689 stars in the catalogue, providing slight redundancy for star identification, and the proposed method can provide an accuracy of at least 1 arc second for an SSR, while also providing robust uniformity and completeness.

In summary, GSC generation is vital for star identification, and is also vital for attitude determination. Meanwhile, machine learning methods have incomparable advantages for data processing. The KNN classification algorithm was shown to be better than the SVM in generating a GSC. Furthermore, we added a new accuracy criterion to the generation standard of the GSC. This method is suitable for high-accuracy attitude determination.

## Figures and Tables

**Figure 1 sensors-21-02647-f001:**
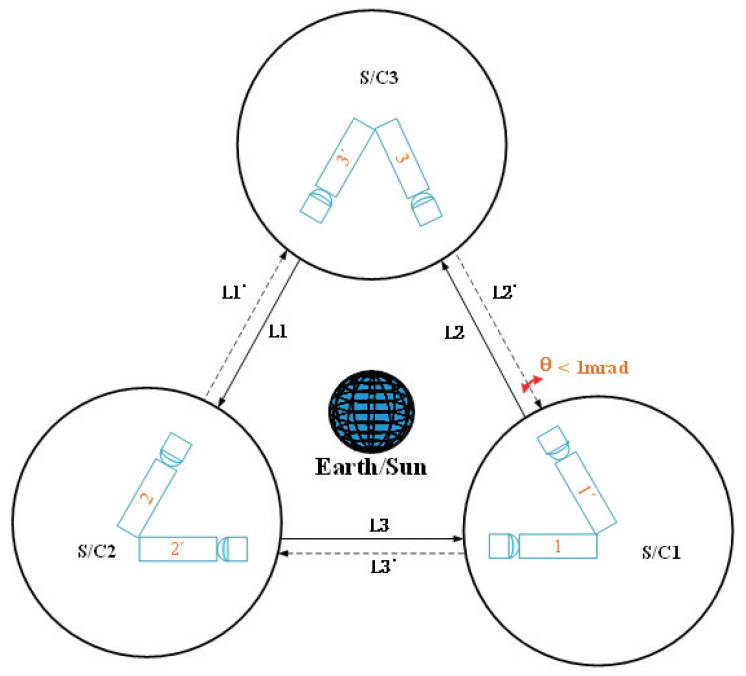
Laser link acquisition for gravitational wave detection.

**Figure 2 sensors-21-02647-f002:**
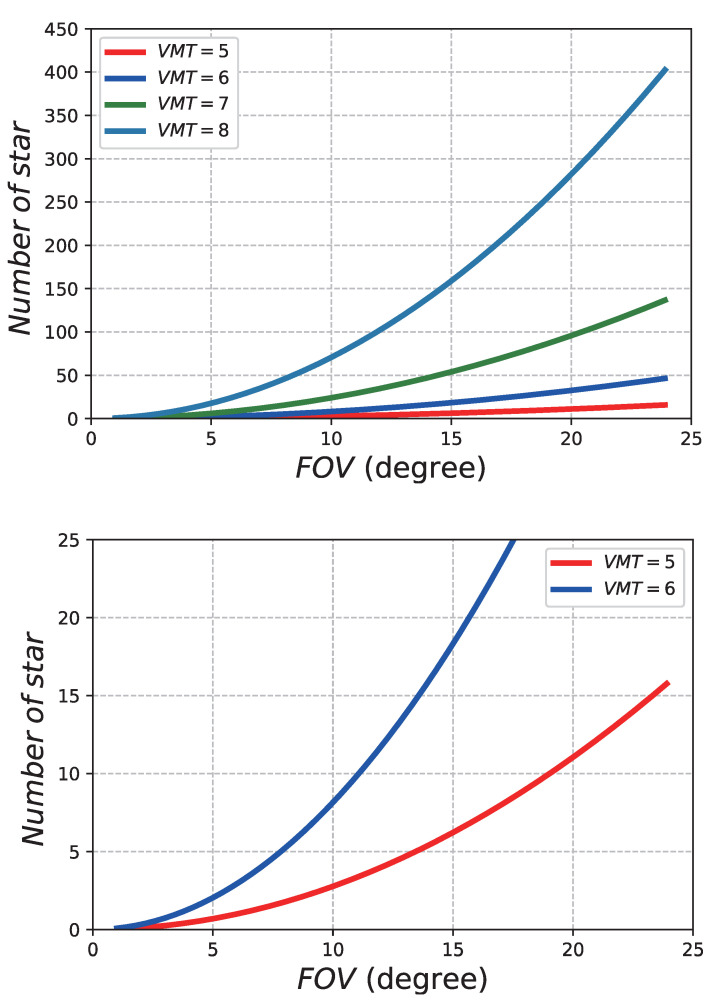
Average number of stars (NOS) for Different fields of view (FOVs) and visual magnitude thresholds (VMTs).

**Figure 3 sensors-21-02647-f003:**
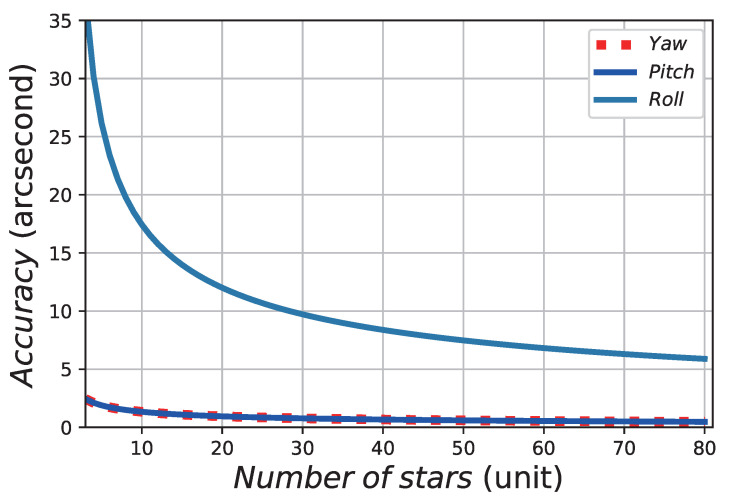
The relationship between the accuracy of a star sensor (SSR) and the NOS in the FOV.

**Figure 4 sensors-21-02647-f004:**
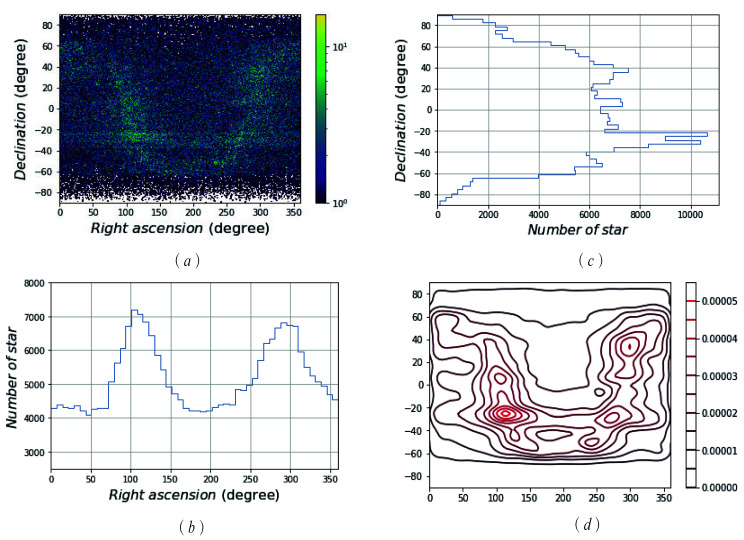
The scattered data density of the SA0 star catalogue. (**a**) The distribution density of SAO GSC; (**b**) the statistics of the NOS distributed in the direction of right ascension; (**c**) the statistics of the NOS distributed in the direction of declination; (**d**) the nuclear density line of SAO GSC stars in the celestial sphere.

**Figure 5 sensors-21-02647-f005:**
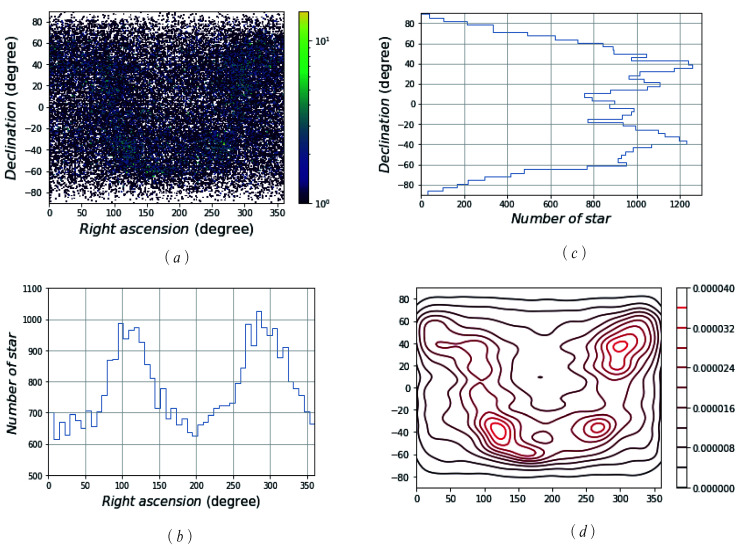
The scattered data density of the MFM star catalogue. (**a**) The distribution density of MFM GSC; (**b**) the statistics of the NOS distributed in the direction of right ascension; (**c**) the statistics of the NOS distributed in the direction of declination; (**d**) the nuclear density line of MFM GSC stars in the celestial sphere.

**Figure 6 sensors-21-02647-f006:**
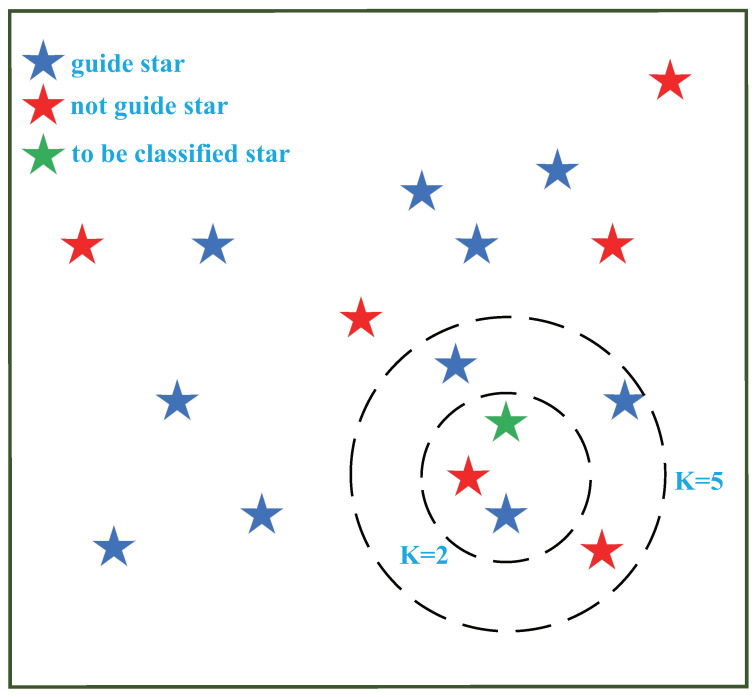
A schematic diagram of the KNN classification algorithm.

**Figure 7 sensors-21-02647-f007:**
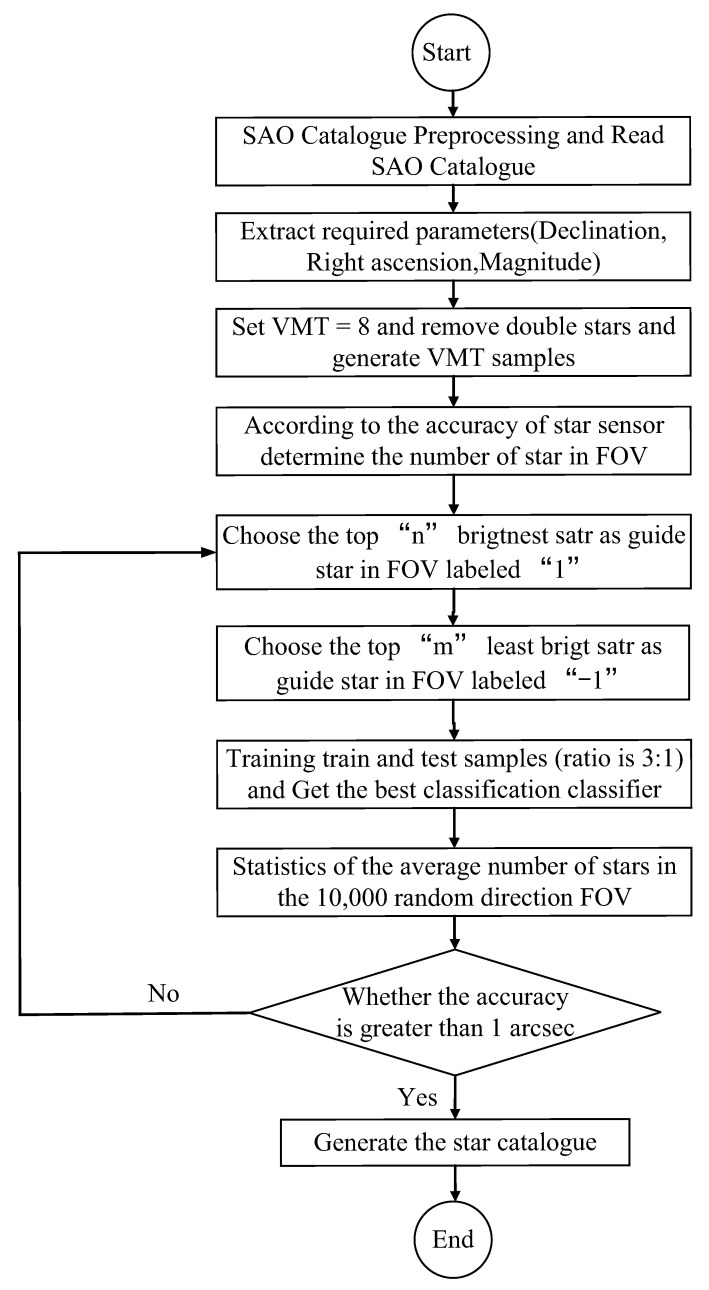
Flowchart of the algorithm used to generate star catalogues.

**Figure 8 sensors-21-02647-f008:**
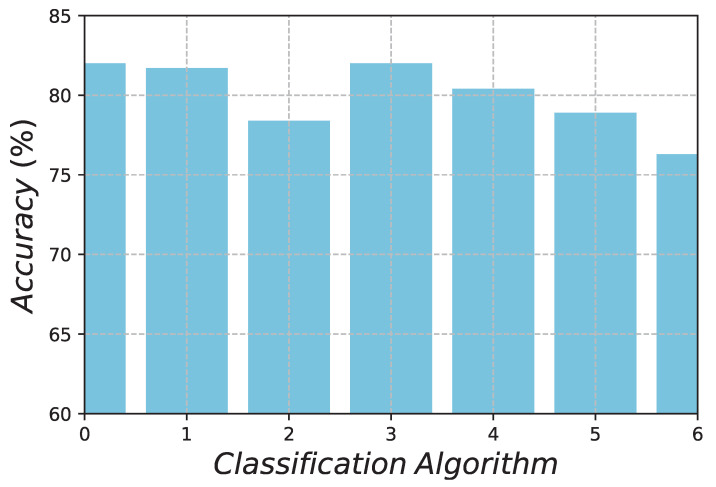
Accuracies of the classification algorithms.

**Figure 9 sensors-21-02647-f009:**
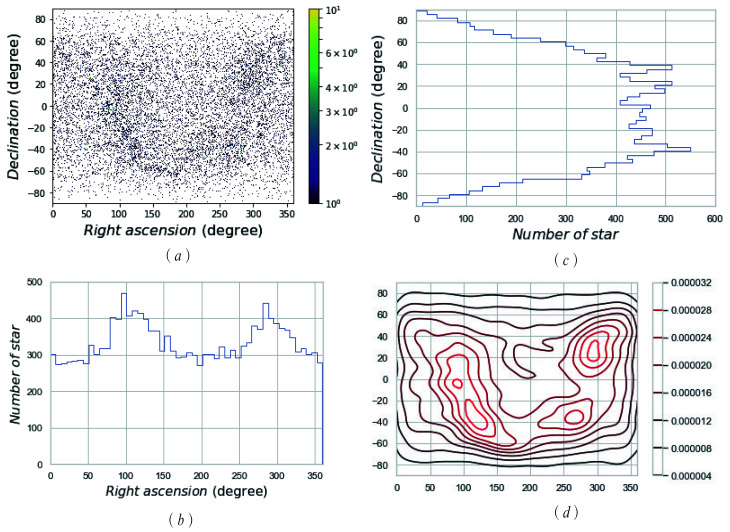
Uniformity and density of the K-Nearest Neighbours (KNN)-generated star catalogue. (**a**) The distribution density of KNN GSC; (**b**) the statistics of the NOS distributed in the direction of right ascension; (**c**) the statistics of the NOS distributed in the direction of declination; (**d**) the nuclear density line of KNN GSC stars in the celestial sphere.

**Table 1 sensors-21-02647-t001:** Full-sky catalogues.

Star Catalogues	Visual Magnitude Threshold	Number of Stars	Epoch
LAL	9.0	47,390	–
Tycho2	14.0	2.5 Million	–
Bright Star Catalogue (BSG)	6.5	9110	J2000.0
Guide Star Catalogue (GSC)	16.0	nearly 20 million	J2000.0
Henry Draper Catalogue (HD)	10.0	359,083	J1900.0
Fifth Fundamental Catalogue (FK5)	9.0	4652	–
Smithsonian Astrophysical Observatory (SAO)	11.0	258,997	J2000.0

**Table 2 sensors-21-02647-t002:** Statistics of different guide star catalogues (GSCs) for a $12^{\circ }\times 12^{\circ }$ FOV, tested in 10,000 random boresight directions.

Classifier	Total Guide Stars	Min	Max	Mean	Std	$n\leq 16\;(1.05^{\prime \prime })$	$n\leq 17\;(1.02^{\prime \prime })$	n≥ 18 (<1${}^{\prime \prime }$)
MFM	38,562	40	337	106.79	41.09	0	0	10,000
SVM	18,627	15	163	50.92	19.38	1	1	9999
DC	19,315	19	163	53.18	20.34	0	0	10,000
GBC	19,315	19	163	53.18	20.34	0	0	10,000
KNN	16,689	15	143	44.73	15.62	3	3	9997
NN	38,562	40	337	106.73	41.09	0	0	10,000
RF	19,315	19	163	53.18	20.34	0	0	10,000
SGD	27,190	19	253	75.50	29.29	0	0	10,000

## Data Availability

Not applicable.

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
