# Peer review of "High-Accuracy Guide Star Catalogue Generation with a Machine Learning Classification Algorithm"

_sensors, 2021, doi:10.3390/s21082647_

Round 1

Reviewer 1 Report

Review of the article
Jianming Zhang, Junxiang Lian, Zhaoxiang, Shuwang Yang and Ying Shan
«High Accuracy Guide Star Catalogue Generation with
Machine Learning Classification Algorithm»

The article is interesting, in my mind, but difficult to read. In my opinion, it requires improvement in several directions. Below are my comments.

  1. English is not my native language, usually I do not evaluate the level of the authors' language, but this text clearly requires language editing. In some places, it is difficult to understand what the authors wanted to say.
  2. The authors a priori require from readers of this article to have a serious knowledge of the Machine Learning Theory. Explanations on many aspects of the machine learning algorithms used by the authors are minimal, the reader should know this or read the original work (not always meaningful).
  3. The authors compared various machine learning methods for constructing GSC, but did not compare these methods with finite and deterministic algorithms for GSC creating. Moreover, the best catalogs created in the study contain 17-19 thousand stars. The computational complexity of creating a GSC of such a volume by deterministic methods is not large, and catalogs can be of higher quality.
  4. There are many problems with references. Reference numbers in the text are out of order. There are no references to works [7,8,33] in the text of the article at all. Some links are completely useless ([28-31]). There are many incomplete references: without the year of publication ([9,14,19,21]), with incomplete author's name ([12,13,15]), with the wrong year ([11]).

Many links to old articles, pre-1990. Several links to articles in Chinese.

  1. The text of the article contains several fragments not related to the main content of the publication. This is the application of star trackers in gravitational wave astronomy on pages 1 and 2 of the Introduction.

The second such segment is devoted to star trackers with three fields of view on page 3.

  1. There are many misprints in the text and figures. I list some below, but I probably did not find all.

The rest of the comments are less essential.

  1. Lines 114-115 contain abbreviations MWM and S-OA, which are not explained and are not found anywhere else.
  2. “Slove”, “sloving” in lines 117-118 is probably misprints.
  3. Line 155 says that angle A is in degrees, but in equation (1) it is in radians.
  4. Why is the factor e^1.08M different from 10^0.4M in equation (1)?
  5. Line 158. There is no proportionality between NFOV and FOV and VMT.
  6. In what units is FOV = 10 given on line 159? The question arose because the expression "12°´12° FOV" is quoted at the end of this paragraph.
  7. Line 162. This is the last line of the paragraph after equation (1), which deals with a circular field of view, but a FOV of "12°´12°" is square.
  8. Figure 2. In the legend writes “VMT” and “VTM”.
  9. Line 164. "Papaer" is a misprint.
  10. If equation (3) contains Nstar – 1, then equation (2) should be as well.
  11. Table 1 and lines 201-212. A very strange choice of catalog for creating GSC. All the GSCs I know for stellar trackers were made from Tycho-2 catalog, and in the last few years from the Gaia DR2 and eDR3 catalogs, which are more complete and have a significantly higher astrometric accuracy than the SAO catalog.
  12. Equations (4) and (5) are not correct.
  13. Bar 219. The equations for alpha0 beta0 and are valid only for double stars of equal brightness.
  14. In the captions to Figures 4 and 5 and in Fig. 7 (in the figure itself) there is a misprint "satr".
  15. Pictures 4, 5, 8. What value does the colors in the upper pictures represent? What value of this quantity correspond to the color in Figures 5 and 8? What does white color?

What do the colors in the pictures below mean? (They do not have a color scale.) In what units are the histograms at the top and right of the figure plotted?

All three figures need to be redone and made more detailed captions.

  1. Line 238 says "star 238 catalog showed in Figure 8 is much better than it in the Figure 5". It is not visible. The lower parts of the figures differ in color, and the upper parts in density to the points (although they should differ in colors), but the meaning of these differences is not clear from the text. See previous note.
  2. Figure 6. Accuracy of the classification algorithms ranges from 82% to 77%. This is a very narrow interval. Are these differences statistically significant? (There are no error bars on the histogram.) What does Accuracy = 82% mean? This is 82% of what?
  3. It is not clear from Figure 7 why the catalog build cycle should ever end? In the loop, 10,000 FOVs are randomly selected to check the accuracy of the catalog, but all other actions are deterministic.
  4. Table 2.

1) What values are shown in the three right-hand columns of the table?

2) Judging by the fact that the values for Total, Min and Max are integers, each method created only one GSC. Why was this done? It would be better to create several catalogs using different random samples and average the results.

  1. Line 270 says "distribution of stars in the catalog is unknown". This is not true, the distribution of stars in any catalog up to a given VMT value is known with very high accuracy.
  2. Line 272. The authors acknowledge the best KNN method and call it "novel", but refer to its description in the 1951 publication [27].
  3. Lines 274-278. It is impossible to understand from the text what exactly was done in articles [28-31].
  4. Lines 281 and 282. What does the value of w_j mean?
  5. Line 315. “Mente Carlo” is misprint.

Author Response

On behalf of my co-authors, we are very grateful to your comments for the manuscript. According with your advice, we amended the relevant part in manuscript. we appreciate you for your very detailed and insightful review report again.

Please see the attachment for your reply!

Reviewer 2 Report

This paper aims to generate a uniform and complete star catalogue by using the classification algorithm called K-Nearest Neighbors (KNN).

The content of the paper is not correctly organized. As examples, algorithm are introduced in section 5 Discussion; Figure 7 with the algorithm flow chart is not referenced in the text.

How the KNN is applied is totally unclear. A more detailed description of input, output data and learning parameters is quite necessary.

The benefit of the proposed algorithm is not well addressed, as there are many parameters that are not studied and it leads to a no fair comparison.

It is necessary to fix numerous typo errors and unmeaningful sentences and to review uncomplete references to facilitate the bibliography search.

Author Response

On behalf of my co-authors, we appreciate you very much for your careful review and constructive suggestions with regard to our manuscript “High Accuracy Guide Star Catalogue Generation with Machine Learning Classification Algorithm”. 

Please see the attachment for your reply!

Round 2

Reviewer 1 Report

The authors corrected almost all of my comments, for the rest they gave explanations with which I agreed. The necessary additions have been made to the text of the article.
I believe that after this the article may be accept in present form.

Author Response

Dear Reviewer,

On behalf of my co-authors, we are very grateful to your affirmation of this article.  We thank you again for your detailed and profound comments. These do help a lot in improving the quality of the draft. We will continue to improve the quality of this manuscript.

Thanks again!

Best wishes for you!

Jianming Zhang, Junxiang Lian, Zhaoxiang Yi, et al.

Reviewer 2 Report

a) The readiness of the paper has been significantly improved. This new version is well organized, and a detailed explanation of the algorithm has been included.

b) The following paragraph shall be clarified:

We chose “n”, the brightest star in each simulated FOV, as a guide star and labelled it as “1”, meaning that we first traversed the entire sky, according to the right ascension and declination. Then, we took half of the FOV to simulate SSR imaging. Later, we chose “m”, the least bright star, as a guide star and labelled it as “-1”. The stars “n” and “m”, taken together, form a sample set with labels.

b.1) It seems that “n” is not a start, but a set with the brightest stars in a given FoV, because you choose n=12. The same occurs with m.

b.2) Why do you label the “m” least bright stars with “-1”? it is mentioned that you choose them as guide stars, as the previous “n”.

b.3) In the proposed algorithm, the least bright stars are chosen as guide stars. However, these least bright stars will suffer from noise more than the brightest stars when calculating its centroid. Please, include a comment about this.

c) The spelling and grammar can be improved, especially when writing “brigt” instead of “bright”.

Author Response

Dear reviewer,

On behalf of my co-authors, we are very grateful to your comments for the manuscript. According with your advice, we amended the relevant part in manuscript. we appreciate you for your very detailed and insightful review report again. We answer your comments in the attachment.

Your question is really due to my wrong expression. I apologize again!
